# LEARNING MODELS FOR VISUAL 3D LOCALIZATION WITH IMPLICIT MAPPING

## ABSTRACT

We consider learning based methods for visual localization that do not require the construction of explicit maps in the form of point clouds or voxels. The goal is to learn an implicit representation of the environment at a higher, more abstract level, for instance that of objects. We propose to use a generative approach based on Generative Query Networks (GQNs, Eslami et al., 2018), asking the following questions: 1) Can GQN capture more complex scenes than those it was originally demonstrated on? 2) Can GQN be used for localization in those scenes? To study this approach we consider procedurally generated Minecraft worlds, for which we can generate images of complex 3D scenes along with camera pose coordinates. We first show that GQNs, enhanced with a novel attention mechanism can capture the structure of 3D scenes in Minecraft, as evidenced by their samples. We then apply the models to the localization problem, comparing the results to a discriminative baseline, and comparing the ways each approach captures the task uncertainty.

## 1 INTRODUCTION

The problem of identifying the position of the camera that captured an image of a scene has been studied extensively for decades (Triggs et al., 1999; Thrun et al., 2005; Cadena et al., 2016). With applications in domains such as robotics and autonomous driving, a considerable amount of engineering effort has been dedicated to developing systems for different versions of the problem. One formulation, often referred to as simply 'localization', assumes that a map of the 3D scene is provided in advance and the goal is to localize any new image of the scene relative to this map. A second formulation, commonly referred to as 'Simultaneous Localization and Mapping' (SLAM), assumes that there is no prespecified map of the scene, and that it should be estimated concurrently with the locations of each observed image. Research on this topic has focused on different aspects and challenges including: estimating the displacement between frames in small time scales, correcting accumulated drifts in large time scales (also known as 'loop closure'), extracting and tracking features from the observed images (e.g. Lowe, 2004; Mur-Artal et al., 2015), reducing computational costs of inference (graph-based SLAM, Grisetti et al., 2010) and more.

Although this field has seen huge progress, performance is still limited by several factors (Cadena et al., 2016). One key limitation stems from reliance on hand-engineered representations of various components of the problem (e.g. key-point descriptors as representations of images and occupancy grids as representations of the map), and it has been argued that moving towards systems that operate using more abstract representations is likely to be beneficial (Salas-Moreno et al., 2013; McCormac et al., 2017). Considering the map, it is typically specified as either part of the input provided to the system (in localization) or part of its expected output (in SLAM). This forces algorithm designers to define its structure explicitly in advance, e.g. either as 3D positions of key-points, or an occupancy grid of a pre-specified resolution. It is not always clear what the optimal representation is, and the need to pre-define this structure is restricting and can lead to sub-optimal performance.

In this work, we investigate the problem of localization with *implicit* mapping, by considering the 're-localization' task, using learned models with no explicit form of map. Given a collection of 'context' images with known camera poses, 're-localization' is defined as finding the relative camera pose of a new image which we call the 'target'. This task can also be viewed as loop closure without an explicit map. We consider deep models that learn implicit representations of the map, where at

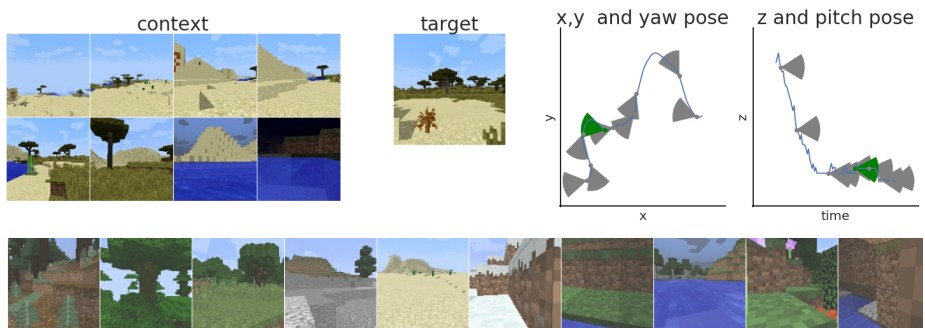

Figure 1: The Minecraft random walk dataset for localization in 3D scenes. We generate random trajectories in the Minecraft environment, and collect images along the trajectory labelled by camera pose coordinates (x,y,z yaw and pitch). Bottom: Images from random scenes. Top: The localization problem setting - for a new trajectory in a new scene, given a set of images and corresponding camera poses (the 'context'), predict the camera pose of an additional observed image (the 'target', in green).

the price of making the map less interpretable, the models can capture abstract descriptions of the environment and exploit abstract cues for the localization problem.

In recent years, several methods for localization that are based on machine learning have been proposed. Those methods include models that are trained for navigation tasks that require localization. In some cases the models do not deal with the localization explicitly, and show that implicit localization and mapping can emerge using learning approaches (Mirowski et al., 2016; Cueva & Wei, 2018; Banino et al., 2018; Wayne et al., 2018). In other cases, the models are equipped with specially designed spatial memories, that can be thought of as explicit maps (Zhang et al., 2017; Gupta et al., 2017a; Parisotto & Salakhutdinov, 2017). Other types of methods tackle the localization problem more directly using models that are either trained to output the relative pose between two images (Zamir et al., 2016; Ummenhofer et al., 2017), or models like PoseNet (Kendall et al., 2015), trained for re-localization, where the model outputs the camera pose of a new image, given a context of images with known poses. Models like PoseNet do not have an explicit map, and therefore have an opportunity to learn more abstract mapping cues that can lead to better performance. However, while machine learning approaches outperform hand-crafted methods in many computer vision tasks, they are still far behind in visual localization. A recent study (Walch et al., 2017) shows that a method based on traditional structure-from-motion approaches (Sattler et al., 2017a) significantly outperforms PoseNet. One potential reason for this, is that most machine learning methods for localization, like PoseNet, are discriminative, i.e. they are trained to directly output the camera pose. In contrast, hand-crafted methods are usually constructed in a generative approach, where the model encodes geometrical assumptions in the causal direction, concerning the reconstruction of the images given the poses of the camera and some key-points. Since localization relies heavily on reasoning with uncertainty, modeling the data in the generative direction could be key in capturing the true uncertainty in the task. Therefore, we investigate here the use of a generative learning approach, where on one hand the data is modeled in the same direction as hand-crafted methods, and on the other hand there is no explicit map structure, and the model can learn implicit and abstract representations.

A recent generative model that has shown promise in learning representations for 3D scene structure is the Generative Query Network (GQN) (Eslami et al., 2018). The GQN is conditioned on different views of a 3D scene, and trained to generate images from new views of the same scene. Although the model demonstrates generalization of the representation and rendering capabilities to held-out 3D scenes, the experiments are in simple environments that only exhibit relatively well-defined, small-scale structure, e.g. a few objects in a room, or small mazes with a few rooms. In this paper, we ask two questions: 1) Can the GQN scale to more complex, visually rich environments? 2) Can the model be used for localization? To this end, we propose a dataset of random walks in the game Minecraft (figure 1), using the Malmo platform (Johnson et al., 2016). We show that coupled with a novel attention mechanism, the GQN can capture the 3D structure of complex Minecraft scenes, and can use this for accurate localization. In order to analyze the advantages and shortcomings of the model, we compare to a discriminative version of it, similar to previously proposed methods.

In summary, our contributions are the following: 1) We suggest a generative approach to localization with implicit mapping and propose a dataset for it. 2) We enhance the GQN using a novel sequential attention model, showing it can capture the complexity of 3D scenes in Minecraft. 3) We show that GQN can be used for localization, finding that it performs comparably to our discriminative baseline when used for point estimates, and investigate how the uncertainty is captured in each approach.

## 2 DATA FOR LOCALIZATION WITH IMPLICIT MAPPING

In order to study machine learning approaches to visual localization with implicit mapping, we create a dataset of random walks in a visually rich simulated environment. We use the Minecraft environment through the open source Malmo platform (Johnson et al., 2016). Minecraft is a popular computer game where a player can wander in a virtual world, and the Malmo platform allows us to connect to the game engine, control the player and record data. Since the environment is procedurally generated, we can generate sequences of images in an unlimited number of different scenes. Figure 1 shows the visual richness of the environment and the diversity of the scene types including forests, lakes, deserts, mountains etc. The images are not realistic but nevertheless contain many details such as trees, leaves, grass, clouds, and also different lighting conditions.

Our dataset consists of 6000 sequences of 100 images each generated by a blind exploration policy based on simple heuristics (see appendix A for details). We record images at a resolution of $128 \times 128$ although for all the experiments in this paper we downscale to $32 \times 32$. Each image is recorded along with a 5 dimensional vector of camera pose coordinates consisting of the $x, y, z$ position and yaw and pitch angles (omitting the roll as it is constant). Figure 1 also demonstrates the localization task: For every sequence we are given a context of images along with their camera poses, and an additional target image with unknown pose. The task is to find the camera pose of the target image.

While other datasets for localization with more realistic images have been recently proposed (Armeni et al., 2016; Savva et al., 2017; Chang et al., 2017; Song et al., 2017), we choose to use Minecraft data for several reasons. First, in terms of the visual complexity, it is a smaller step to take from the original GQN data, which still poses interesting questions on the model's ability to scale. The generative approach is known to be hard to scale but nevertheless has advantages and this intermediate dataset allows us to demonstrate them. The second reason, is that Minecraft consists of outdoors scenes which compared to indoor environments, contain a 3D structure that is larger in scale, and where localization should rely on more diverse clues such as using both distant key-points (e.g. mountains in the horizon) and closer structure (e.g. trees and bushes). Indoor datasets also tend to be oriented towards sequential navigation tasks relying much on strong priors of motion. Although in practice, agents moving in an environment usually have access to dense observations, and can exploit the fact that the motion between subsequent observations is very small, we focus here on localization with a sparse observation set, since we are interested in testing the model's ability to capture the underlying structure of the scene only from observations and without any prior on the camera's motion. One example for this setting is when multiple agents can broadcast observations to help each other localize. In this case there are usually constraints on the network bandwidth and therefore on the number of observations, and there is also no strong prior on the camera poses which all come from different cameras.

## 3 MODEL

The localization problem can be cast as an inference task in the probabilistic graphical model presented in figure 2a. Given an unobserved environment $E$ (which can also be thought of as the map), any observed image $X$ depends on the environment $E$ and on the pose $P$ of the camera that captured the image $Pr(X|P, E)$. The camera pose $P$ depends on the environment and perhaps some prior, e.g. a noisy odometry sensor. In this framing, localization is an inference task involving the posterior probability of the camera pose which can be computed using Bayes' rule:

$$Pr(P|X, E) = \frac{1}{Z} Pr(X|P, E) Pr(P|E) \tag{1}$$

In our case, the environment is only implicitly observed through the context, a set of image and camera pose pairs, which we denote by $C = \{x_i, p_i\}$. We can use the context to estimate the

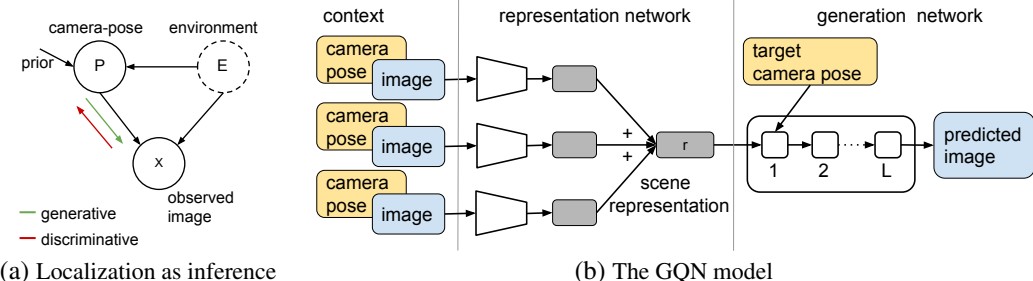

(a) Localization as inference  (b) The GQN model

Figure 2: Localization as probabilistic inference (a). The observed images $X$ depend on the environment $E$ and the camera pose $P$. To predict $Pr(P|X)$, in the generative approach, we use a model of $Pr(X|P)$ (green) and apply Bayes' rule. In the discriminative approach we directly train a model of $Pr(P|X)$ (red). In both cases the environment is implicitly modeled given a context of (image, camera pose) pairs $C = \{x_i, p_i\}$. In the GQN model (b), the context is processed by a representation network and the resulting scene representation is fed to a recurrent generation network with L layers, predicting an image given a camera pose.

environment $\hat{E}(C)$ and use the estimate for inference:

$$Pr(P|X, C) = \frac{1}{Z} Pr(X|P, \hat{E}(C)) Pr(P|\hat{E}(C)) \tag{2}$$

Relying on the context to estimate the environment $E$ can be seen as an empirical Bayes approach, where the prior for the data is estimated (in a point-wise manner) at test time from a set of observations. In order to model the empirical Bayes likelihood function $Pr(X|P, \hat{E}(C))$ we consider the Generative Query Network (GQN) model (Eslami et al., 2018) (figure 2b). We use the same model as described in the GQN paper, which has two components: a representation network, and a generation network. The representation network processes each context image along with its corresponding camera pose coordinates using a 6-layer convolutional neural network. The camera pose coordinates are combined to each image by concatenating a 7 dimensional vector of $x, y, z, sin(yaw), cos(yaw), sin(pitch), cos(pitch)$ to the features of the neural network in the middle layer. The output of the network for each image is then added up resulting in a single scene representation. Conditioned on this scene representation and the camera pose of a new target view, the generation network which consists of a conditional latent-variable model DRAW (Eslami et al., 2018) with 8 recurrent layers, generates a probability distribution of the target image $Pr_{GQN}(X|P, C)$. We use this to model $Pr(X|P, \hat{E}(C))$, where the scene representation $r$ serves as the point-wise estimate of the environment $\hat{E}(C)$.

Given a pre-trained GQN model as a likelihood function, and a prior over the camera pose $Pr(P|C)$, localization can be done by maximizing the posterior probability of equation 2 (MAP inference):

$$\arg\max_P Pr(P|X, C) = \arg\max_P \log Pr_{GQN}(X|P, C) + \log P(P|C) \tag{3}$$

With no prior, we can simply resort to maximum likelihood, omitting the second term above. The optimization problem can be a hard problem in itself, and although we show some results for simple cases, this is usually the main limitation of this generative approach.

### 3.1 ATTENTION

One limitation of the GQN model is the fact that the representation network compresses all the information from the context images and poses to a single global representation vector $r$. Although the GQN has shown the ability to capture the 3D structure of scenes and generate samples of new views that are almost indistinguishable from ground truth, it has done so in fairly simple scenes with only a few objects in each scene and limited variability. Since we are interested here in more complex scenes with much more visual variability, it is unlikely for a simple representation network with a fixed size scene representation to be able to retain all the important information of the scene. Following previous work on using attention for making conditional predictions (Vinyals et al., 2016;

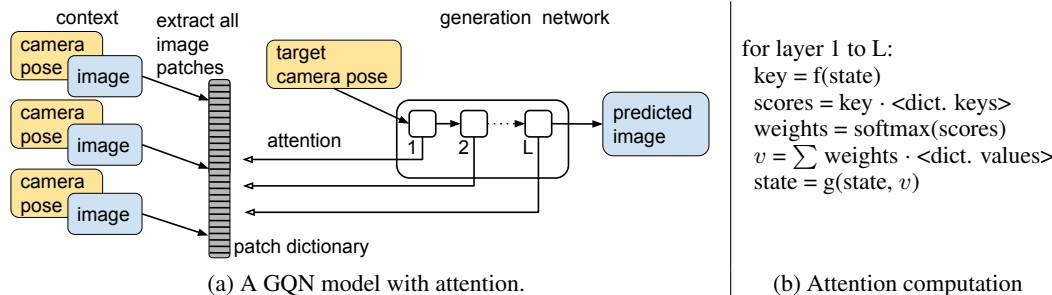

Figure 3: (a) Instead of conditioning on a parametric representation of the scene, all patches from context images are stored in a dictionary, and each layer of the generation network can access them through an attention mechanism. In each layer, the key is computed as a function of the generation network's recurrent state, and the result $v$ is fed back to the layer (b).

Reed et al., 2017; Parisotto et al., 2018), we propose a variation of the GQN model that relies on attention rather than a parametric representation of the scene.

We enhance the GQN model using an attention mechanism on patches inspired by (Reed et al., 2017) (figure 3), where instead of passing the context images through a representation network, we extract from each of the images all the 8 by 8 patches and concatenate to each the camera poses and the 2D coordinates of the patch within the image. For each patch we also compute a key using a convolutional neural network, and place all patches and their corresponding keys in a large dictionary, which is passed to the generation network. In the generation network, which is based on the DRAW model with 8 recurrent layers, a key is computed at every layer from the current state, and used to perform soft attention on the patch dictionary. The attention is performed using a dot-product with all the patches' keys, normalizing the results to one using a softmax, and using them as the weights in a weighted sum of all patches. The resulting vector is then used by the generation network in the same way as the global representation vector is used in the parametric GQN model. See appendix B for more details.

In this sequential attention mechanism, the key in each layer depends on the result of the attention at the previous layer, which allows the model to use a more sophisticated attention strategy, depending both on the initial query and on patches and coordinates that previous layers attended to.

## 3.2 A DISCRIMINATIVE APPROACH

As a baseline for the proposed generative approach based on GQN, we use a discriminative model which we base on a 'reversed-GQN'. This makes the baseline similar to previously proposed methods (see section 4), but also similar to the proposed GQN model in terms of architecture and the information it has access to, allowing us to make more informative comparisons.

While the generative approach is to learn models in the $P \rightarrow X$ direction and invert them using Bayes' rule as described above, a discriminative approach would be to directly learn models in the $X \rightarrow P$ direction. In order to model $Pr(P|X,C)$ directly, we construct a 'reversed-GQN' model, by using the same GQN model described above, but 'reversing' the generation network, i.e. using a model that is conditioned on the target image, and outputs a distribution over the camera pose. The output of the model is a set of categorical distributions over the camera pose space, allowing multi-modal distributions to be captured. In order to keep the output dimension manageable, we split the camera pose coordinate space to four components: 1) the $x, y$ positions, 2) the $z$ position, 3) the yaw angle, and 4) the pitch angle. To implement this we process the target image using a convolutional neural network, where the scene representation is concatenated to the middle layer, and compute each probability map using an MLP.

We also implement an attention based version of this model, where the decoder's MLP is preceded by a sequential attention mechanism with 10 layers, comparable to the attention in the generative model. For more details on all models see appendix B.

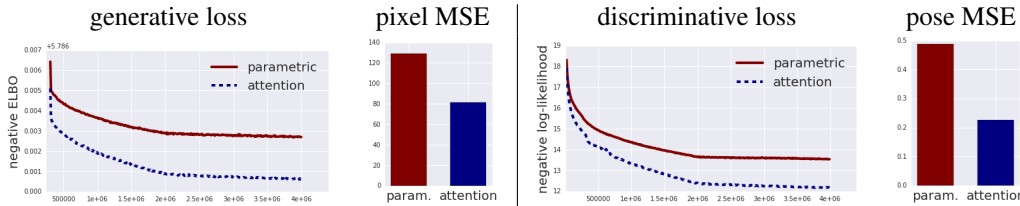

Figure 4: The loss and predictive MSE for both the generative direction and the discriminative direction. The attention model results in lower loss and lower MSE in both directions.

Although the discriminative approach is a more direct way to solve the problem and will typically result in much faster inference at test time, the generative approach has some aspects that can prove advantageous: it learns a model in the causal direction, which might be easier to capture, easier to interpret and easier to break into modular components; and it learns a more general problem not specific to any particular task and free from any prior on the solution, such as the statistics that generated the trajectories. In this way it can be used for tasks which it was not trained for, for example different types of trajectories, or for predicting the x,y location in a different spatial resolution.

## 4 RELATED WORK

In recent years, machine learning has been increasingly used in localization and SLAM methods. In one type of methods, a model is trained to perform some navigation task that requires localization, and constructed using some specially designed spatial memory that encourages the model to encode the camera location explicitly (Zhang et al., 2017; Parisotto & Salakhutdinov, 2017; Gupta et al., 2017a;b; Henriques & Vedaldi, 2018; Savinov et al., 2018; Parisotto et al., 2018). The spatial memory can then be used to extract the location. Other models are directly trained to estimate either relative pose between image pairs (Zamir et al., 2016; Ummenhofer et al., 2017), or camera re-localization in a given scene (Kendall et al., 2015; Walch et al., 2017; Clark et al., 2017).

All these methods are discriminative, i.e. they are trained to estimate the location (or a task that requires a location estimate) in an end-to-end fashion, essentially modeling $Pr(P|X)$ in the graphical model of figure 2. In that sense, our baseline model using a reversed-GQN, is similiar to the learned camera re-localization methods like PoseNet (Kendall et al., 2015), and in fact can be thought of as an adaptive (or meta-learned) PoseNet, where the model is not trained on one specific scene, but rather inferred at test time, thus adapting to different scenes. The reason we implement an adaptive PoseNet rather than comparing to the original version is that in our setting it is unlikely for PoseNet to successfully learn the network's weights from scratch for every scene, which contains only a small number of images. In addition, having a baseline which is as close as possible to the proposed model makes the direct comparison of the approaches easier.

In contrast to the more standard discriminative approach, the model that we propose using GQN, is a generative approach that captures $Pr(X|P)$, making it closer to traditional hand-crafted localization methods that are usually constructed with geometric rules in the generative direction. For example, typical methods based on structure-from-motion (SfM) (Li et al., 2012; Zeisl et al., 2015; Svärm et al., 2017; Sattler et al., 2017a) optimize a loss involving reconstruction in image space given pose estimates (the same way GQN is trained).

The question of explicit vs. implicit mapping in hand-crafted localization methods, has been studied in (Sattler et al., 2017b), where it was shown that re-localization relying on image matching can outperform methods that need to build complete 3D models of a scene. With learned models, the hope is that we can get even more benefit from implicit mapping since it allows the model to learn abstract mapping cues that are hard to define in advance. Two recent papers, Bloesch et al. (2018); Tang & Tan (2018), take a somewhat similar approach to ours by encoding depth information into an implicit encoding, and training the models based on image reprojection error. Other recent related work (Zhou et al., 2017; Tulsiani et al., 2018) take a generative approach to pose estimation but explicitly model depth maps or 3D shapes.

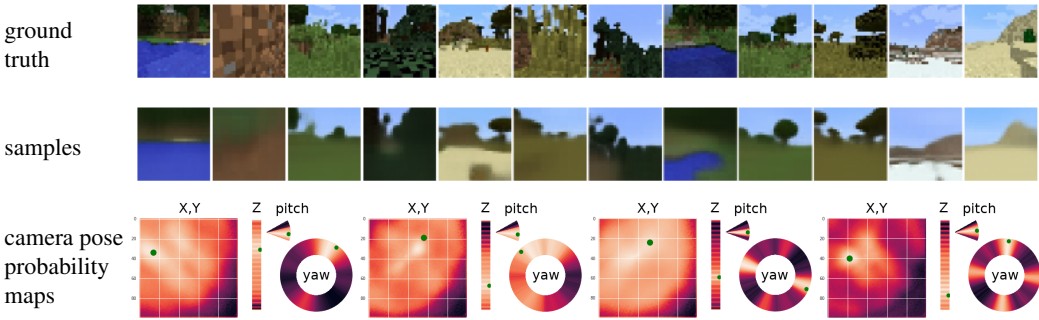

ground truth

samples

camera pose probability maps

Figure 5: Generated samples from the generative model (middle), and the whole output distribution for the discriminative model (bottom). Both were computed using the attention models, and each image and pose map is from a different scene conditioned on 20 context images. The samples capture much of the structure of the scenes including the shape of mountains, the location of lakes, the presence of large trees etc. The distribution of camera pose computed with the reversed GQN shows the model's uncertainty, and the distribution's mode is usually close to the ground truth (green dot).

## 5 TRAINING RESULTS

We train the models on the Minecraft random walk data, where each sample consists of 20 context images and 1 target image, from a random scene. The images are drawn randomly from the sequence (containing 100 images) in order to reduce the effect of the prior on the target camera pose due to the sequential nature of the captured images. The loss we optimize in the generative direction is the negative variational lower bound on the log-likelihood over the target image, since the GQN contains a DRAW model with latent variables. For the discriminative direction with the reversed GQN which is fully deterministic, we minimize the negative log-likelihood over the target camera pose.

Figure 4 shows the training curves for the generative and discriminative directions. In both cases we compare between the standard parametric model and the attention model. We also compare the MSE of the trained models, computed in the generative direction by the L2 distance between the predicted image and the ground-truth image, and in the discriminative direction by the L2 distance between the most likely camera pose and the ground truth one. All results are computed on a held out test set containing scenes that were not used in training, and are similar to results on the training set. A notable result is that the attention models improve the performance significantly. This is true for both the loss and the predictive MSE, and for both the discriminative and generative directions.

Figure 5 shows generated image samples from the generative model, and the predicted distribution over the camera pose space from the reversed GQN model. All results are from attention based models, and each image and camera pose map comes from a different scene from a held out test set, conditioned on 20 context images. Image samples are blurrier than the ground truth but they capture the underlying structure of the scene, including the shape of mountains, the location of lakes, the presence of big trees etc. For the reversed GQN model, the predicted distribution shows the uncertainty of the model, however the mode of the distribution is usually close to the ground truth camera pose. The results show that some aspects of the camera pose are easier than others. Namely, predicting the height $z$ and the pitch angle are much easier than the $x, y$ location and yaw angle. At least for the pitch angle this can be explained by the fact that it can be estimated from the target image alone, not relying on the context. As for the yaw angle, we see a frequent pattern of high probability areas around multiples of 90 degrees. This is due to the cubic nature of the Minecraft graphics, allowing the estimate of the yaw angle up to any 90 degrees rotation given the target image only. This last phenomena is a typical one when comparing discriminative methods to generative ones, where the former is capable of exploiting shortcuts in the data.

One interesting feature of models with attention, is that they can be analyzed by observing where they are attending, shedding light on some aspects of the way they work. Figure 6 shows the attention in three different scenes. Each row shows 1) the target image, 2) the x,y trajectory containing the poses of all context images (in black) and the pose of the target image (in green), and 3) the context images with a red overlay showing the patches with high attention weights. In each row we show the context

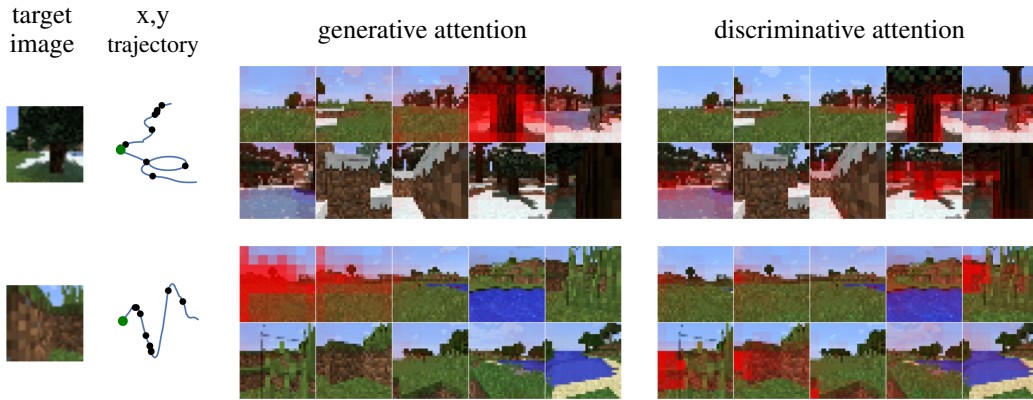

Figure 6: Attention over the context images in the generative and discriminative directions. The total attention weights are shown as a red overlay, using the same context images for both directions. Similar to hand-crafted feature point extraction and graph-SLAM, the learned attention is sparse and prunes out irrelevant images. While the generative attention is mainly position based, focusing on one or two of the nearest context images, the discriminative attention is based more on appearance, searching all context images for patches resembling the target. See also supplementary video.

images twice, once with the attention weights of the generative model, and once with the attention weights of the discriminative model. In both cases we show the total attention weights summed over all recurrent layers. A first point to note is that the attention weights are sparse, focusing on a small number of patches in a small number of images. The attention focuses on patches with high contrast like the edges between the sky and the ground. In these aspects, the model has learned to behave similarly to the typical components of hand crafted localization methods - detecting informative feature points (e.g. Harris corner detection), and constructing a sparse computation graph by pruning out the irrelevant images (e.g. graph-SLAM). A second point to note is the difference between the generative and discriminative attention, where the first concentrates on fewer images, covering more space within them, and the second is distributed between more images. Since the generative model is queried using a camera pose, and the discriminative model using an image, it is perhaps not surprising that the resulting attention strategies tend to be position-based and appearance-based respectively. For more examples see the video available at `https://youtu.be/iHEXX5wXbCI`.

## 6    LOCALIZATION

We compare the generative and discriminative models ability to perform localization, i.e. find the camera pose of a target image, given a context of image and camera pose pairs. For the discriminative models we simply run a forward pass and get the probability map of the target's camera pose. For the generative model, we fix the target image and optimize the likelihood by searching over the camera pose used to query the model. We do this for both the x,y position, and the yaw, using grid search with the same grid values as the probability maps that are predicted by the discriminative model. The optimization of the x,y values, and the yaw values is done separately while fixing all other dimensions of the camera pose to the ground truth values. We do this without using a prior on the camera pose, essentially implementing maximum likelihood inference rather than MAP.

Figure 7 shows the results for a few held-out scenes comparing the output of the discriminative model, with the probability map computed with the generative model. For both directions we use the best trained models which are the ones with attention. We see that in most cases the maximum likelihood estimate (magenta star) is a good predictor of the ground truth value (green circle) for both models. However it is interesting to see the difference in the probability maps of the generative and discriminative directions. One aspect of this difference is that while in the discriminative direction the x,y values near the context points tend to have high probability, for the generative model it is the opposite. This can be explained by the fact that the discriminative model captures the posterior of the camera pose, which also includes the prior, consisting of the assumption that the target image was taken near context images. The generative model only captures the likelihood $Pr(X|P)$, and

Table 1: Localization with a varying number of context images. Using attention ('+att') and more context images leads to better results. Even with no prior on poses, the generative+attention model results in better MSE. However the discriminative+attention model assigns higher probability to the ground-truth pose since the prior allows it to concentrate more mass near context poses.

MSE

| context | x,y position | | | yaw angle | | |
|---|---|---|---|---|---|---|
| | 5 | 10 | 20 | 5 | 10 | 20 |
| disc | 0.27 | 0.25 | 0.27 | 0.25 | 0.21 | 0.17 |
| gen | 0.41 | 0.36 | 0.35 | 0.16 | 0.14 | 0.15 |
| disc+att | 0.17 | 0.12 | 0.09 | 0.21 | 0.15 | 0.11 |
| gen+att | **0.11** | **0.08** | **0.07** | **0.06** | **0.06** | **0.05** |

log-probability

| | x,y position | | | yaw angle | | |
|---|---|---|---|---|---|---|
| | 5 | 10 | 20 | 5 | 10 | 20 |
| | -8.14 | -7.24 | -7.08 | -3.27 | -3.08 | -3.02 |
| | -10.19 | -9.08 | -8.55 | -5.95 | -5.16 | -4.74 |
| | -7.65 | **-6.47** | **-5.74** | **-3.56** | **-3.29** | **-3.08** |
| | **-7.56** | -6.62 | -6.00 | -4.48 | -4.06 | -3.66 |

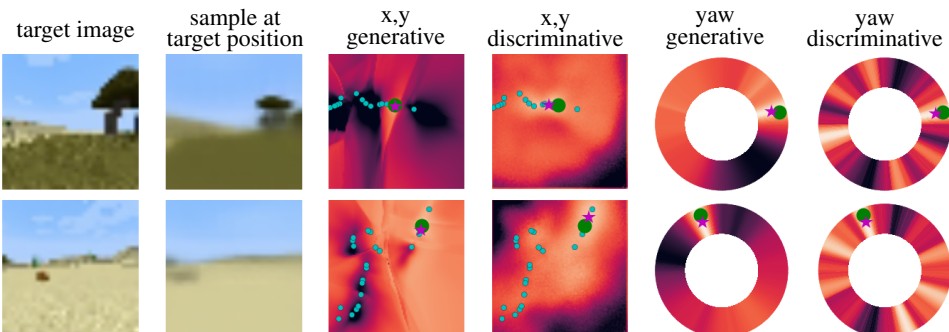

Figure 7: Localization with the generative and discriminative models - target image, a sample drawn using the ground-truth pose, and probability maps for x,y position and yaw angle (bright=high prob.). The generative maps are computed by querying the model with all possible pose coordinates, and the discriminative maps are simply the model's output. The poses of the context images are shown in cyan, target image in green and maximum likelihood estimates in magenta. The generative maps are free from the prior on poses in the training data giving higher probability to unexplored areas.

therefore is free from this prior, resulting in an opposite effect where the area near context images tend to have very low probability and a new image is more likely to be taken from an unexplored location. See appendix figure 9 for more examples.

Table 1 shows a quantitative comparison of models with and without attention, with varying number of context points. The first table compares the MSE of the maximum likelihood estimate of the x,y position and the yaw angle, by computing the square distance from the ground truth values (again, for the generative model we fix all other dimensions of the pose to the ground truth values). The table shows the improvement due to attention, and that as expected, more context images result in better estimates. We also see that the generative models estimates have a lower MSE for all context sizes. This is true even though the generative model does not capture the prior on camera poses and can therefore make very big mistakes by estimating the target pose far from the context poses. The second table compares the log probability of the ground truth location for the x,y position and yaw angles (by using the value in the probability map cell which is closest to the ground truth). Here we see that the discriminative model results in higher probability. This can be caused again by the fact that the generative model does not capture the prior of the data and therefore distributes a lot of probability mass across all of the grid. We validate this claim by computing the log probability in the vicinity of all context poses (summing the probability of a 3 by 3 grid around each context point) and indeed get a log probability of -4.1 for the generative direction compared to -1.9 for the discriminative one (using attention and 20 context images). Another support for the claim that the discriminative model relies more on the prior of poses rather than the context images themselves is that in both tables, the difference due to attention is bigger for the generative model compared to the discriminative model.

## 7 DISCUSSION

We have proposed a formulation of the localization problem that does not involve an explicit map, where we can learn models with implicit mapping in order to capture higher level abstractions. We have enhanced the GQN model with a novel attention mechanism, and showed its ability to capture the underlying structure of complex 3D scenes demonstrated by generating samples of new views of the scene. We showed that GQN-like models can be used for localization in a generative and discriminative approach, and described the advantages and disadvantages of each.

When comparing the approaches for localization we have seen that the generative model is better in capturing the underlying uncertainty of the problem and is free from the prior on the pose that is induced by the training data. However, the clear disadvantage of the generative approach is that it requires performing optimization at test time, while the discriminative model can be used by running one forward pass. We believe that future research should focus on combining the approaches which can be done in many different ways, e.g. using the discriminative model as a proposal distribution for importance sampling, or accelerating the optimization using amortized inference by training an inference model on top of the generative model (Rosenbaum & Weiss, 2015). While learning based methods, as presented here and in prior work, have shown the ability to preform localization, they still fall far behind hand-crafted methods in most cases. We believe that further development of generative methods, which bear some of the advantages of hand-crafted methods, could prove key in bridging this gap, and coupled with an increasing availability of real data could lead to better methods also for real-world applications.

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

## A    THE MINECRAFT RANDOM WALK DATASET FOR LOCALIZATION

To generate the data we use the Malmo platform (Johnson et al., 2016), an open source project allowing access to the Minecraft engine. Our dataset consists of 6000 sequences of random walks consisting of 100 images each. We use 1000 sequences as a held-out test set. The sequences are generated using a simple heuristic-based blind policy as follows:

We generate a new world and a random initial position, and wait for the agent to drop to the ground.

For 100 steps:

1. Make a small random rotation and walk forward.
2. With some small probability make a bigger rotation.
3. If no motion is detected jump and walk forward, or make a bigger rotation
4. Record image and camera pose.

We prune out sequences where there was no large displacement or where all images look the same (e.g. when starting in the middle of the ocean, or when quickly getting stuck in a hole), however in some cases our exploration policy results in close up images of walls, or even underwater images which might make the localization challenging.

We use 5 dimensional camera poses consisting of $x, y, z$ position and yaw and pitch angles (omitting the roll as it is constant). We record images in a resolution of $128 \times 128$ although for all the experiments in this paper we downscale to $32 \times 32$. We normalize the x,y,z position such that most scenes contain values between -1 and 1.

## B    MODEL DETAILS

The basic model that we use is the Generative Query Network (GQN) as described in (Eslami et al., 2018). In the representation network, every context image is processed by a 6 layer convolutional neural network (CNN) outputting a representation with spatial dimension of $8 \times 8$ and $64$ channels. We add the camera pose information of each image after 3 layers of the CNN by broadcasting the values of $x, y, z, sin(yaw), cos(yaw), sin(pitch), cos(pitch)$ to the whole spatial dimension and concatenate as additional 7 channels. The output of each image is added up to a single scene representation of $8 \times 8 \times 64$. The architecture of the CNN we use is: [k=2,s=2,c=32] $\rightarrow$ [k=3,s=1,c=32] $\rightarrow$ [k=2,s=2,c=64] $\rightarrow$ [k=3,s=1,c=32] $\rightarrow$ [k=3,s=1,c=32] $\rightarrow$ [k=3,s=1,c=64], where for each layer 'k' stands for the kernel size, 's' for the stride, and 'c' for the number of output channels.

In the generation network we use the recurrent DRAW model as described in (Eslami et al., 2018) with 8 recurrent layers and representation dimension of $8 \times 8 \times 128$ (used as both the recurrent state and canvas dimensions). The model is conditioned on both the scene representation and the camera pose query by injecting them (using addition) to the biases of the LSTM's in every layer. The output of the generation network is a normal distribution with a fixed standard deviation of $0.3$.

In order to train the model, we optimize the negative variational lower bound on the log-likelihood, using Adam (Kingma & Ba, 2014) with a batch size of 36, where each example comes from a random scene in Minecraft, and contains 20 context images and 1 target image. We train the model for 4M iterations, and anneal the output standard deviation starting from $1.5$ in the first iteration, down to $0.3$ in the 300K'th iteration, keeping it constant in further iterations.

### B.1    REVERSED GQN

The reversed GQN model we use as a discriminative model is based on the GQN described above. We use the same representation network, and a new decoder that we call the localization network, which is queried using the target image, and outputs a distribution of camera poses (see figure 8). In order to make the output space manageable, we divide it to 4 log-probability maps:

1. A matrix over $x, y$ values between -1 and 1, quantized in a $0.02$ resolution.
2. A vector over $z$ values between -1 and 1, quantized in a $0.02$ resolution.

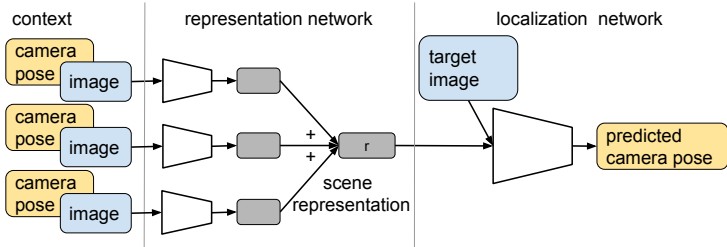

Figure 8: The reversed GQN model. A similar architecture to GQN, where the generation network is 'reversed' to form a localization network, which is queried using a target image, and predicts its camera pose.

3. A vector over yaw values between -180 and 180 degrees, quantized in a 1 degree resolution.

4. A vector over pitch values between -20 and 30 degrees, quantized in a 1 degree resolution.

The localization network first processes the image query using a CNN with the following specifications: [k=3,s=2,c=32] $\to$ [k=3,s=2,c=64] $\to$ [k=3,s=1,c=64] $\to$ [k=1,s=1,c=64] $\to$ [k=1,s=1,c=64], resulting in an intermediate representation of $8 \times 8 \times 64$. Then it concatenates the scene representation computed by the representation network and processes the result using another CNN with: [k=3,s=1,c=64] $\to$ [k=3,s=1,c=64] $\to$ [k=3,s=1,c=64] $\to$ [k=5,s=1,c=4]. Each of the 4 channels of the last layer is used to generate one of the 4 log-probability maps using an MLP with 3 layers, and a log-softmax normalizer. Figure 5 in the main paper shows examples of the resulting maps.

## B.2 ATTENTION GQN

In the attention GQN model, instead of a scene encoder as described above, all $8 \times 8 \times 3$ patches are extracted from the context images with an overlap of 4 pixels and placed in a patch dictionary, along with the camera pose coordinates, the 2D patch coordinates within the image (the x,y, position in image space), and a corresponding key. The keys are computed by running a CNN on each image resulting in a $8 \times 8 \times 64$ feature map, and extracting the (channel-wise) vector corresponding to each pixel. The architecture of the CNN is [k=2,s=2,c=32] $\to$ [k=3,s=1,c=32] $\to$ [k=2,s=2,c=64] $\to$ [k=1,s=1,c=32] $\to$ [k=1,s=1,c=32] $\to$ [k=1,s=1,c=64], such that every pixel in the output feature map corresponds to a $4 \times 4$ shift in the original image. The keys are also concatenated to the patches.

Given the patch dictionary, we use an attention mechanism within the recurrent layers of the generation network based on DRAW as follows. In each layer we use the recurrent state to compute a key using a CNN of [k=1,s=1,c=64] $\to$ [k=1,s=1,c=64] followed by spatial average pooling. The key is used to query the dictionary by computing the dot product with all dictionary keys, normalizing the results using a softmax. The normalized dot products, are used as weights in a weighted sum over all dictionary patches and keys (using the keys as additional values). See pseudo-code in figure 3b. The result is injected to the computation of each layer in the same way as the scene representation is injected in the standard GQN. Since the state in each layer of the recurrent DRAW depends on the computation of the previous layer, the attention becomes sequential, with multiple attention keys where each key depends on the result of the attention with the previous key.

In order to implement attention in the discriminative model, and make it comparable to the sequential attention in DRAW, we implement a similar recurrent network used only for the attention. This is done by adding 10 recurrent layers between layer 5 and 6 of the localization network's CNN. Like in DRAW, a key is computed using a 2 layer CNN of [k=1,s=1,c=64] $\to$ [k=1,s=1,c=64] followed by spatial average pooling. The key is used for attention in the same way as in DRAW, and the result is processed through a 3 layer MLP with 64 channels and concatenated to the recurrent state. This allows the discriminative model to also use a sequential attention strategy where keys depend on the attention in previous layers.

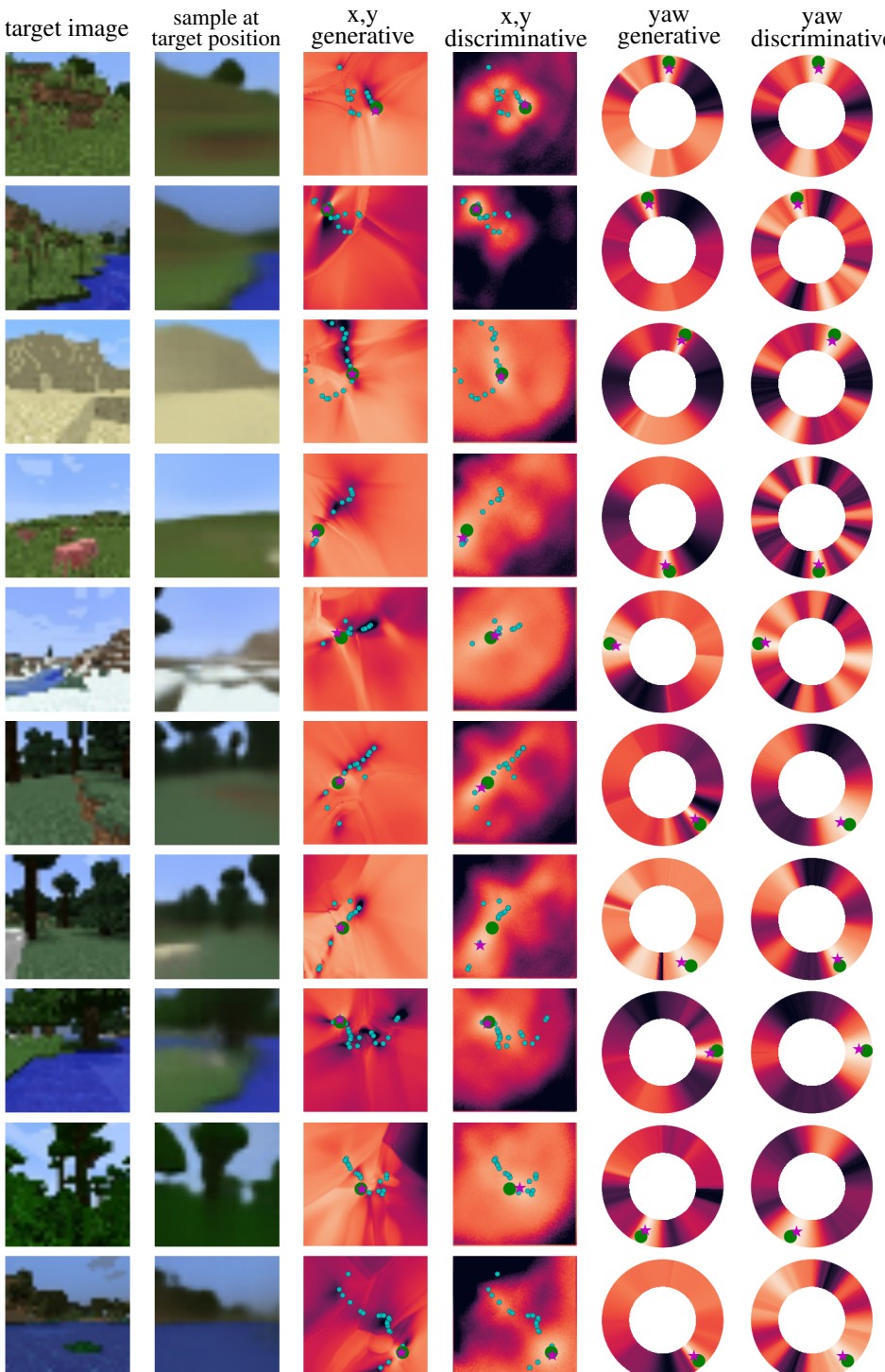

Figure 9: Localization with the generative and discriminative models - target image, a sample drawn using the ground-truth pose, and probability maps for the x,y position and yaw angles. The generative maps are computed by querying the model with all possible pose coordinates, and the discriminative maps are simply the model's output. The poses of the context images are shown in cyan, target image in green and maximum likelihood estimates in magenta. The generative maps are free from the prior in the training data (that target poses are close to the context poses) giving higher probability to unexplored areas and lower probabilities to areas with context images which are dissimilar to the target.

