# OpenReview forum: "Learning models for visual 3D localization with implicit mapping"
_ICLR.cc/2019/Conference_

### Official Review · AnonReviewer3 · 2018-10-31
**An interesting but incremental application of Eslami et al. (2018)**

**Rating:** 6
**Confidence:** 3

**Review:**

Summary:
Eslami et al. (2018) proposed a deep neuronal framework for a scene representation and renderer (the Generative Query Networks: GQN), which generate an image from a scene representation and a query camera pose. In this work, the authors use the GQN to estimate the camera pose from a target image. Existing learning approaches are discriminative, meaning that they are trained to output the camera pose in an end-to-end fashion, while this paper proposes a generative method more in the line of hand-crafted methods which still largely outperform learning approaches. Using the GQN with the proposed attention mechanism, the method captures an implicit mapping of the environment at a more abstract level. This implicit representation is then used to optimize the likelihood of the target pose in a probabilistic graphical model framing. They compare their solution to a discriminative baseline, based on a reversed GQN.

Pros:
- As shown in Figure 7, the generative approach seems to capture better the implicit representation associated to the mapping from the scene geometry and the image.
- The proposed generative solution seems to be more accurate than the discriminative baseline.
- As shown in Table 1, the proposed attention mechanism allows to focus on relevant parts of the context images, giving flexibility for more complex scenes.
- Unlike classical discriminative methods, the proposed solution can be easily used in new scenes (different from the one used for the learning) thanks to the representation network.

Cons:
- The contribution seems incremental with respect to Eslami et al. (2018).
- Lack of comparisons to state of the art, in particular a comparison with PoseNet is necessary.
- The results are shown only on simple datasets of small images (32x32 pixels).
- Tradeoff between precision and time computing is necessary to handle large environments because of space discretization. Then, the method seems to be far to be exploited in a real life SLAM application (e.g. autonomous vehicle).

---

> ### Author Response · Authors · 2018-11-22
> **Thank you for your review and helpful comments.**
>
> Thank you for your review and helpful comments.
>
> Regarding using PoseNet as a baseline - As we describe briefly in the related work, Our discriminative baseline, the reversed-GQN,  can be thought of as an implementation of an adaptive PoseNet, where instead of training the weights from scratch on new scenes,  we train a representation network that outputs a scene-specific vector on which the decoder is conditioned. The reason we need to make PoseNet adaptive is because the number of observations in our task is likely to be too small to train a whole network from scratch. Additionally, the fact that the architecture of our baseline and the input it is conditioned on are similar to our proposed generative model, reduces confounding factors and makes the comparison of the approaches easier. We have emphasizes this further in section 4.
>
> Regarding the work being incremental - The main objective and the novelty of this paper is to propose a generative approach and demonstrate the advantage and disadvantage compared to the discriminative approach which is the standard in recent work.
> However, even when considering the model itself, we think that the novel sequential attention mechanism that we introduced is a significant and interesting extension of the recent GQN model. This is what enables the model to move from toy data with a handful of simple objects in a room, to the more complex scenes in Minecraft (see figure 4 for quantitative comparison), and this is also what allows the model to be used for localization (see table 1).  The visualization of the attention (figure 6) is also an interesting outcome when comparing to the classic methods of image feature extraction for localization.
>
> Regarding using more complex data and real life applications -  We agree that demonstrating our method on real data with more complex scenes and higher resolution images would be interesting, however since we are proposing a new approach and we base it on a model that was previously only shown to work on toy data,  we think that it is important to take intermediate steps and analyze the results. As far as we know,  all the localization methods based on machine learning still fall far behind handcrafted methods and therefore are still in an exploratory phase. Furthermore, within the machine learning methods, as we mention in the paper,  the generative approach that we propose has a clear limitation of computation time (since it needs to optimize at test time). Using real data would make this disadvantage even worse, potentially preventing us from demonstrating the benefits of the approach (better capturing the uncertainty and being free from a prior), which we think are important to the machine learning community and could drive further progress. We have emphasized this point further in section 2.

---

> > ### Comment · AnonReviewer3 · 2018-11-26
> > **Reviewer 3 additional comments**
> >
> > We thank the authors for the provided additional details. After reading their responses, I upgrade my initial rating of 5 to 6.

---

### Official Review · AnonReviewer2 · 2018-11-02
**Weak evaluation**

**Rating:** 5
**Confidence:** 4

**Review:**

**Summary of the paper**

This paper studies the problem of visual re-localization, where we are interested in estimating the camera pose of a new image from a set of source images and their camera poses. Instead of explicitly designing the structure of a map of 3D scenes (e.g. occupancy grids or point clouds), the paper proposes implicitly learning an abstract map representation. Specifically, the paper proposes a generative method based on Generative Query Networks (GQNs) augmented with an attention mechanism. The authors apply this model to the visual re-localization problem. To train and test the proposed model, the authors introduce the Minecraft random walk dataset, which consists of images and their camera poses extracted from randomly generated trajectories in the Minecraft environment. The proposed model is compared against a discriminative counterpart, which is trained to directly predict the target camera pose and achieves better MSE.

**Clarity**

Above average

**Significance**

Below Average

**Detailed comments**

_Paper Strengths_

- The idea of leveraging generative models' knowledge of "maps" to perform visual localization is interesting. This gives learning frameworks the flexibility of building a latent representaiton of maps which may yield better performance instead of being restricted to a pre-defined representations.
- The paper is very well-written and easy to follow.
- The authors did a good job presenting the proposed methods. The descriptions and formulations are clear. Both Figure 2 and Figure 3 are helpful for understanding the GQNs and the proposed attention mechanism.
- The patch dictionary for the attention mechanism seems effective especially when dealing with a set of context images capturing the same scene.
- The authors are honest about the limitations of the proposed framework compared to classic approaches.
- The visualization of results are clear. Particularly, Figure 5 and Figure 7 give easily interpretable representations of the results.

_Paper Weaknesses_

- Implicitly learning a map of the scene is mentioned as a strength in the paper, but this comes at the high cost of interpretability. Without an explicit map representation, it is difficult to understand the failure cases - does the model not understand the 3D scene well or does the model have a hard time accurately predicting camera poses?
- Minecraft is an interesting environment for proof of concept, but lacks much of the subtlety of the real world.
- Building a framework that is able to perform the localization task from real-world scenes is more interesting. Learning generative models of real-world scenes is known to be difficult, which makes this framework impractical. There are google streetview and indoor datasets authors can try to utilize.
- The aforementioned point is supported by the fact that the localization performance of the proposed model on real-world scenes is missing.
- The reviewer does not find enough novelty from the proposed model, which is an iterative improvement on GQNs.
- The paper only compares the proposed model against its discriminative couterpart, which is not sufficient. While the authors strongly argue that exploiting the proposed implicitly representations of scenes is more beneficial than utilizing the pre-defined explicit representations, the only baseline is using the same implicit representations. Although the reviewer is aware of that this model does not use complete video sequences, benchmarking against a visual monocular SLAM algorithm, like LSD-SLAM [1], would contextualize the claim.
- Why quantize the discriminative model's output? This de-correlates nearby pose values. The paper could benefit from an explanation of not using a straightforward regression over pose variables.
- Overall, the reviewer does not find enough novelty from any aspects except the idea of utilizing a generative model for visual localization with implocitly learned maps, which is not fully demonstrated in the experiment section (i.e. not compare to baselines using explicit maps).
- A differentiating factor for this paper could be tackling one of the open problems remaining in SLAM as identified in [2], like lifelong learning or semantic mapping.


_Reproducibility_

- Given the clear description in the main paper and the details provided in the appendix, the reviewer believes reproducing the results is possible.

_Conclusion_

- Overall, the reviewer believes this paper is well presented and reproducible. However, the paper does not propose to solve a novel problem, nor does it present a very novel method. Although the idea of using existing generative networks for localization is interesting, the paper misses important baselines relying on explicit map representation and is not sufficiently convincing. Moreover, requiring a generative model significantly limits the possibility of utilizing the proposed model for real-world applications. While the paper does present a new dataset built in Minecraft which is suitable for demonstrating the strengths of the proposed method, the reviewer does not find this significant. Therefore, the reviewer recommends a rejection.

_Reference_

[1] Engel, Jakob, Thomas Schöps, and Daniel Cremers. "LSD-SLAM: Large-scale direct monocular SLAM." European Conference on Computer Vision. Springer, Cham, 2014.
[2] Cadena, Cesar, et al. "Past, present, and future of simultaneous localization and mapping: Toward the robust-perception age." IEEE Transactions on Robotics 32.6 (2016): 1309-1332.

---

> ### Author Response · Authors · 2018-11-22
> **Thank you for your review and helpful comments.**
>
> Thank you for your review and helpful comments.
>
> Regarding the interpretability of implicit mapping models - We agree that the model is less interpretable in this aspect, and we emphasized this further in the introduction.  However one approach that we find useful for understanding how the model captures the scene structure, is to look at samples from new view points (figures 5, 7, 9) and compare to the ground truth image. More broadly, although neural representations have demonstrated their efficacy on a wide range of tasks, interpretation of neural representations remains an active area of research. We believe many of the findings in interpretability will transfer to our model as well.
>
> Regarding the quantization of the discriminative model’s output - We quantized the output because we wanted to capture the full distribution in the simplest way. This is because we are interested in analyzing the way the (potentially multi-modal) uncertainty is captured. In addition, quantizing the output with the same values as the ones we use for the grid search we perform with the generative approach makes the comparison easier.
> In contrast to other ways of capturing multi-modal distributions (like mixture models), the downside of quantization is the de-correlation of nearby values and that the smoothness needs to be learned from data. However, looking at the output maps compared to the generative localization maps (figure 7 and 9) we see that the discriminative output is much more smooth, and in fact we mention this over-smoothing around the context points as one of the problems of the discriminative method.
>
> Regarding the novelty in the paper - The main novelty is in the approach itself. The main objective of this paper is to propose the generative approach and demonstrate the advantage and disadvantage compared to the discriminative approach which is the standard in recent work.
> However, even when considering the model itself, we think that the novel sequential attention mechanism that we introduced is a significant and interesting extension of the recent GQN model. This is what enables the model to move from toy data with a handful of simple objects in a room, to the more complex scenes in Minecraft (see figure 4 for quantitative comparison), and this is also what allows the model to be used for localization (see table 1).  The visualization of the attention (figure 6) is also an interesting outcome when comparing to the classic methods of image feature extraction for localization.
>
> Regarding using real data - We agree that demonstrating our method on real data with more complex scenes and higher resolution images would be interesting, however since we are proposing a new approach and we base it on a model that was previously only shown to work on toy data,  we think that it is important to take intermediate steps and analyze the results. As far as we know,  all the localization methods based on machine learning still fall far behind handcrafted methods and therefore are still in an exploratory phase. Furthermore, within the machine learning methods, as we mention in the paper,  the generative approach that we propose has a clear limitation of computation time (since it needs to optimize at test time). Using real data would make this disadvantage even worse, potentially preventing us from demonstrating the benefits of the approach (better capturing the uncertainty and being free from a prior), which we think are important to the machine learning community and could drive further progress. We have emphasized this point further in section 2.
>
> Regarding a comparison to LSD-SLAM or a similar method -  As you mention, applying LSD-SLAM will probably not work because the re-localization task is different than SLAM and specifically contains a sparse set of observations with a weaker sequential prior. However we do believe that it is possible to develop a handcrafted method for our task that will work better than our proposal (e.g. bundle-adjustment on some key-point matches).
> We don’t think that such a comparison will be useful here, because as we mention in the introduction and discussion, handcrafted methods in general still outperform machine learning approaches for localization (see Walch et al. 2017 for example). We do believe that machine learning approaches could eventually outperform handcrafted methods, and we think that a generative approach that better captures the uncertainty like we demonstrate in the paper, could be a key factor in making this happen.

---

> > ### Comment · AnonReviewer2 · 2018-12-12
> > **Additional comment**
> >
> > The authors did not address my concerns much. I keep my rating to 5 (leaning to 4), as I still think there are more experiments that can be included to strengthen the paper (and I think  the paper would benefit from this in a long run).
> >
> > [Comparison to traditional SLAM]
> > While the claimed main contribution of the paper is that utilizing implicit mapping would benefit the model on the localization task, no comparison to a method with explicit mapping is shown. Direct visual SLAM approaches, like LSD-SLAM, should perform well in the Minecraft environment given sequential image input. These methods also offer the additional benefit of interpretable explicit geometric maps. Despite comments, the authors do not offer an experiment comparing their approach to traditional approaches, and therefore we are left in the dark about the relative performance of this approach. I suggest the authors compare with the SLAM methods by using a sequence of frames collected in the Minecraft environment and perform the localization task given sampled frames. Therefore, the strengths of the proposed method and the SLAM methods need to be discussed.
> >
> > [Using real data]
> > We are still far from achieving a generative model for real world scenes yet. Therefore, it is not convincing that the proposed approach will address localization problems with real data. The rebuttal did not address this concern.
> >
> > [Interpretability]
> > The proposed model representation is much less interpretable compared to traditional SLAM algorithms as the map is implicitly represented. The authors do not provide insight addressing this point but simply left this for further research without a clear path towards superiority to traditional SLAM.
> >
> > [Quantization of discriminative model output]
> > I believe de-correlating nearby pose values will hurt the performance of the discriminative model and possibly lead to an incorrect conclusion. However, no further experiment to defend this question was provided during the rebuttal period.

---

### Official Review · AnonReviewer1 · 2018-11-05
**Very informative and great paper**

**Rating:** 7
**Confidence:** 4

**Review:**

This paper proposes generative approaches to localization without explicit high definition geometric maps. A generative baseline (GQN) and an extension to that with attention is introduced in the context of localization.

The paper is clearly written, and the relevant previous work is discussed to satisfying degree.

Figures 2 and 3 help understanding the GQN and the proposed attention version a lot.

I am intrigued with the result presented in table 1. especially with the fact that attention is helping the generative method quite a bit, but not so much for the discriminative method. This is not discussed in detail in the paper, I suggest the authors expand their discussion a bit in this direction.

The second suggestion is in terms of the data. I understand the motivations behind using the minecraft world. however, real data is still quite different than this data, think about various differences in the real world at time of training and test even at the same location texture of sky and lighting will change. On top of this, there is quite a bit of variance in levels of detail compared to the monotonic minecraft world. I suggest using a real dataset for another set of experiments. This can be added as an appendix.

In general, very good motivation, and intriguing work for the community -- localization with high definition geometric works is tough to scale, and implicit world representations are an important piece in relaxing this dependency. I believe this paper will motive more future work.

---

> ### Author Response · Authors · 2018-11-22
> **Thank you for your comments and positive feedback.**
>
> Thank you for your comments and positive feedback.
>
> Regarding the effect of attention in the discriminative model - We agree that it is smaller than in the generative direction, however table 1 shows there’s still a significant difference resulting from the attention (compare line 1 with line 3). This is also visible on the right side of figure 4. We think this might happen because a lot of what the discriminative model captures comes from the prior on camera positions rather than using the images. We added that to our discussion on table 1.
>
> Regarding using real data - We agree that demonstrating our method on real data with more complex scenes and higher resolution images would be interesting, however since we are proposing a new approach and we base it on a model that was previously only shown to work on toy data,  we think that it is important to take intermediate steps and analyze the results. As far as we know,  all the localization methods based on machine learning still fall far behind handcrafted methods and therefore are still in an exploratory phase. Furthermore, within the machine learning methods, as we mention in the paper,  the generative approach that we propose has a clear limitation of computation time (since it needs to optimize at test time). Using real data would make this disadvantage even worse, potentially preventing us from demonstrating the benefits of the approach (better capturing the uncertainty and being free from a prior), which we think are important to the machine learning community and could drive further progress. We have emphasized this point further in section 2.

---

### Public Comment · (anonymous) · 2018-11-19
**Is the number of training iterations correct?**

Thanks for your great work. I have one question about your paper.

In Appendix B, you mention "We train the model for 400M iterations", but I wonder if this is a typo of "400K" or "4M" because it takes too much time to train if it is true (In the work of GQN by Eslami et al., they train the model for 2M iterations).

If this is not a typo, I would like to ask you why this model needs so many iterations to train compared to simple GQN.

Thank you.

---

> ### Author Response · Authors · 2018-11-22
> **typo fixed => training for 4M iterations.**
>
> You are right this was a typo and we corrected it now.
> We are training the models for 4M iterations. This can also be seen in the training curves in figure 4.
> thanks!

---

### Meta-Review · Area_Chair1 · 2018-12-14
**lack of experiments to obvious geometric baselines and previous learning-based methods for localization**

**Confidence:** 5
**Recommendation:** Reject

**Metareview:**

The paper proposes a method that learns mapping implicitly, by using a generative query network of Eslami et al. with an attention mechanism to learn to predict egomotion. The empirical findings is that training for egomotion estimation alongside the generative task of view prediction helps over a discriminative baseline, that does not consoder view prediction. The model is tested in Minecraft environments.
A comparison to some baseline SLAM-like method, e.g., a method based on bundle adjustment, would be important to include despite beliefs of the authors that eventually learning-based methods would win over geometric methods.  For example, potentially environments with changes can be considered, which will cause the geometric method to fail, but the proposed learning-based method to succeed.

Moreover, there are currently learning based methods for the re-localization problem that the paper would be important to compare against (instead of just cite), such as "MapNet: An Allocentric Spatial Memory for Mapping Environments" of Henriques et al.  and "Active Neural Localization" of Chaplot et al. . In particular, Mapnet has a generative interpretation by using cross-convolutions as part of its architecture, which generalize very well, and which consider the geometric formation process. The paper makes a big distinction between generative and discriminative, however the architectural details behind the egomotion estimation network are potentially more or equally important to the loss used. This means, different discriminative networks depending on their architecture may perform very differently. Thus, it would be important to present quantitative results against such methods that use cross-convolutions for egomotion estimation/re-localization.